# Identifying Predictor Variables for a Composite Risk Prediction Tool for Gestational Diabetes and Hypertensive Disorders of Pregnancy: A Modified Delphi Study

**DOI:** 10.3390/healthcare12131361

**Published:** 2024-07-08

**Authors:** Stephanie Cowan, Sarah Lang, Rebecca Goldstein, Joanne Enticott, Frances Taylor, Helena Teede, Lisa J. Moran

**Affiliations:** 1Monash Centre for Health Research and Implementation, School of Clinical Sciences, Monash University, Mulgrave, VIC 3170, Australia; stephanie.cowan@monash.edu (S.C.); sarah.lang1@monash.edu (S.L.); rebecca.goldstein@monash.edu (R.G.); joanne.enticott@monash.edu (J.E.); helena.teede@monash.edu (H.T.); 2Monash Endocrine and Diabetes Units, Monash Health, Clayton, Melbourne, VIC 3168, Australia; 3Victorian Heart Institute, Monash Health, Clayton, Melbourne, VIC 3168, Australia

**Keywords:** pregnancy, pre-eclampsia, diabetes mellitus, pregnancy-induced hypertension, Delphi consensus

## Abstract

A composite cardiometabolic risk prediction tool will support the systematic identification of women at increased cardiometabolic risk during pregnancy to enable early screening and intervention. This study aims to identify and select predictor variables for a composite risk prediction tool for cardiometabolic risk (gestational diabetes mellitus and/or hypertensive disorders of pregnancy) for use in the first trimester. A two-round modified online Delphi study was undertaken. A prior systematic literature review generated fifteen potential predictor variables for inclusion in the tool. Multidisciplinary experts (n = 31) rated the clinical importance of variables in an online survey and nominated additional variables for consideration (Round One). An online meeting (n = 14) was held to deliberate the importance, feasibility and acceptability of collecting variables in early pregnancy. Consensus was reached in a second online survey (Round Two). Overall, 24 variables were considered; 9 were eliminated, and 15 were selected for inclusion in the tool. The final 15 predictor variables related to maternal demographics (age, ethnicity/race), pre-pregnancy history (body mass index, height, history of chronic kidney disease/polycystic ovarian syndrome, family history of diabetes, pre-existing diabetes/hypertension), obstetric history (parity, history of macrosomia/pre-eclampsia/gestational diabetes mellitus), biochemical measures (blood glucose levels), hemodynamic measures (systolic blood pressure). Variables will inform the development of a cardiometabolic risk prediction tool in subsequent research. Evidence-based, clinically relevant and routinely collected variables were selected for a composite cardiometabolic risk prediction tool for early pregnancy.

## 1. Introduction

Cardiometabolic pregnancy complications, including gestational diabetes (GDM) and hypertensive disorders of pregnancy (HDP), can adversely affect maternal health during pregnancy and later in life [1]. The identification of women with multiple risk factors for GDM or HDP early in pregnancy is recommended to enable early diagnosis or preventative intervention [2,3]. Early commencement of low-dose aspirin has been found to reduce the risk of developing pre-eclampsia [4], and structured antenatal diet and physical activity interventions have been associated with a reduced risk of gestational diabetes, hypertensive disorders of pregnancy and total adverse maternal outcomes [5]. Optimizing women’s cardiometabolic health in pregnancy may assist in addressing a sex-specific risk factor for future type two diabetes (T2DM) and cardiovascular disease [1].

Clinical risk prediction tools provide clinicians with a systematic approach to determine when screening or intervention is required [6]. Clinical risk prediction tools use a combination of risk factors or ‘predictor variables’ to estimate an individual’s risk of a specific clinical outcome [6]. Prognostic risk prediction tools estimate whether an individual is likely to develop a disease or outcome, whereas diagnostic tools assess if an outcome has already occurred [7]. Composite risk prediction models assess the risk of more than one outcome and reduce the need for clinicians to use multiple single-outcome risk prediction tools, thereby aiming to maximize resources and increase efficiency [8]. Single-outcome prognostic risk prediction tools exist for GDM and HDP. To our knowledge, there is no composite risk prediction tool for cardiometabolic risk (both GDM and HDP) in pregnancy [9]. A composite risk prediction tool may enable clinicians to use one tool to systematically screen and target early antenatal lifestyle intervention and prophylactic measures towards women at the highest risk of multiple adverse cardiometabolic pregnancy outcomes.

Developing a risk prediction tool is a multi-phase, rigorous process [8]. The selection of predictor variables is a critical first step in developing a risk prediction tool as the choice of predictors can significantly affect the model’s performance and accuracy. Engaging stakeholders early in the tool development process is recommended to ensure the selection of relevant predictor variables, and the utility and credibility of the final tool [8]. The practicalities of using a risk prediction tool (such as required time, staff training and processes) can influence the uptake of a risk prediction tool into practice. For example, the inclusion of uterine artery Doppler measurements or biochemical markers, such as pregnancy-associated plasma protein (PaPP-A) or Placental Growth Factor (PlGF), in early pregnancy pre-eclampsia risk prediction tools can be financially prohibitive and prevent use in low-resource clinical settings [10]. Engaging key stakeholders to select predictor variables that are acceptable and feasible to collect in the first trimester may improve the uptake of the final tool into practice. This study aims to engage experts and consumers to identify and select predictor variables for inclusion in a prognostic composite risk prediction tool to predict the risk of GDM and/or HDP for use in the first trimester.

## 2. Materials and Methods

A modified two-round online (e-Delphi) study (June–August 2022) was conducted, in line with current guidelines on Delphi techniques in health services (Figure 1) [11]. A Delphi study was used to attain expert consensus on predictor variables for inclusion in an antenatal composite cardiometabolic risk prediction tool. A Delphi study provides a structured process to attain expert feedback and consensus on addressing a clinical problem [11]. A modified e-Delphi process, which combines online anonymous surveys with a consensus meeting, was used to enable expert contribution from varied geographic contexts and facilitate group decision-making [11]. 

### 2.1. Identification of Predictor Variables for the Risk Prediction Tool

A steering/research committee was developed, including experts in risk prediction and biostatistics, women’s health and clinicians. A prior literature review by Thong et al. [9] was used as preliminary research for this modified Delphi study. This review provided an updated summary of contemporary risk prediction models for GDM and HDP in women with singleton pregnancies. This review informed the initial list of predictor variables for consideration by the expert panel. The review by Thong et al. [9] included a systematic search (OVID MEDLINE and PubMed), which identified GDM risk prediction models published between 2016 and June 2021, with the search designed to update the 2016 review by Kenelly and McAuliffe [12]. The HDP risk prediction models published between 2018 to June 2021 were reviewed. A search from 2018 onwards was conducted to reflect the updated classification and diagnosis of HDP outlined by the International Society for the Study of Hypertension in Pregnancy [2]. Please refer to the publication by Thong et al. [9] for a detailed overview of the review search strategy, methodology and findings of this review. Appendix A summarize the prediction models identified by Thong et al. [9], which informed the list of predictor variables presented to the expert panel in the *Round One* survey. Previously published reviews [12,13,14] and national and international screening guidelines for GDM and HDP were also reviewed (Appendix A) to ensure relevant predictor variables in existing risk prediction models, and commonly reported risk factors were considered. 

Fifteen shared candidate predictor variables used in both GDM and HDP models were included in the *Round One* survey. The predictor variables for consideration by the panel were chosen, as they were included in both the HDP and the GDM risk prediction models and had shared pathophysiological pathways for GDM and HDP. Predictor variables included demographic variables (age, ethnicity/race), pre-pregnancy history (body mass index, height, history of smoking, pre-existing/chronic hypertension), obstetric history (gestational weight gain, history of pre-eclampsia, parity), maternal biochemical measures (alanine transaminase, blood glucose levels, pregnancy-associated plasma protein-a) and maternal hemodynamic measures (mean arterial pressure, systolic blood pressure, uterine artery pulsatility index). Appendix A outlines the list and definitions of the predictor variables that were provided to the expert panel.

### 2.2. Panel Selection 

Purposive sampling was used to recruit representatives from nine key stakeholder groups, including obstetricians, endocrinologists, midwives, allied health professionals, epidemiologists, biostatisticians, health economists, general practitioners and women with prior GDM and HDP. Refer to Appendix A for an overview of the Delphi process and panelist recruitment. A list of international candidate panelists was created using contacts from established committees servicing research projects supporting women with cardiometabolic pregnancy complications. Academics, clinician researchers and women on existing consumer advisory committees that advocate for women in healthcare research were invited and volunteered to be on the expert panel. Women had received training on participating in research committees and advocating for women’s health in research. Study investigators with relevant expertise who were not directly involved in data collection or analysis were invited to join the panel (HT, LM, JE, RG). Participants were provided with a detailed explanation of the project and the expectations for participating on the panel.

### 2.3. E-Delphi Round One Online Survey 

Sixty-two panelists were invited to participate in *Round One* via email, with a reminder sent after ten and fourteen days. Panelists were presented with fifteen predictor variables and asked to rate the clinical importance of each predictor on a nine-point Likert scale: one was limited importance, and nine was critically important. Importance was defined as the extent to which the predictor is highly valued or necessary (based on clinical relevance and statistical robustness) to predict the risk of GDM and HDP. Panelists could choose ‘not to comment’ if they did not have the relevant expertise. As panelists were required to understand the clinical relevance of predictor variables in predicting cardiometabolic risk, the women with prior GDM and HDP did not participate. 

Pre-specified consensus criteria, as per prior Delphi workshops [15], were used to synthesize the survey results. The outcomes that scored ≥7 by ≥70% of participants and ≤4 by <15% of participants were included. Outcomes were excluded if they received a score of ≥7 by <15% of participants and a score of ≤4 by ≥70% of participants. The outcomes with other score combinations were considered equivocal and were carried over for consideration in *Round Two* [15].

Panelists were provided with twenty-four predictor variables that did not qualify for inclusion in the initial questionnaire, as the variables were not present in both the HDP and the GDM risk prediction models identified by Thong et al. [9]. These additional variables were used frequently in risk prediction models (at least 15% of models) or were included in models that had undergone extensive external validation (Appendix A) [9]. Panelists were encouraged to nominate additional important predictor variables from this list or nominate additional variables (based on expert opinion) for discussion in the consensus meeting and *Round Two* survey.

### 2.4. Consensus Meeting and Round Two Online Survey

All panelists who participated in *Round One* and six women with prior GDM and HDP were invited to attend a virtual consensus meeting. The consensus meeting involved a presentation of the *Round One* findings, small, structured group discussions and the *Round Two* online survey. Appendix A outlines the aims and structure of the consensus meeting.

Experts were presented with the predictor variables classified as ‘included’ and ‘equivocal’ in *Round One*, the results of the survey (presented using median and interquartile range) and additional nominated predictor variables. Experts then participated in small and large group discussions to explore their opinions on the clinical importance, feasibility (the extent to which the predictor variables can be easily and conveniently measured in early pregnancy) and acceptability (the extent to which the predictor variables are suitable and appropriate to collect in early pregnancy) of the variables. Participants were encouraged to combine or modify variables, such as defining measurement units or updating definitions, to ensure the clinical relevance of the variable in predicting cardiometabolic risk. A nominal group technique was used to encourage contributions from all panelists [16]. Group facilitators maintained field notes to document the ideas discussed by panelists. The notes were summarized to contextualize the questionnaire results. 

The *Round Two* online questionnaire was completed using the same nine-point Likert scale as in *Round One*. Panelists considered the clinical importance, alongside the feasibility and acceptability of collecting predictor variables in early pregnancy. A pre-specified consensus criterion was used to reach a consensus. The outcomes with a median score of ≥7 and an IQR of ≤2 were included. All other outcomes were excluded. The *Round Two* survey was completed anonymously by participants.

Panelists were emailed the *Round Two* survey outcomes for further independent comment following the workshop. The research team deliberated the comments before finalizing the list of predictor variables. The final list was presented at a consumer advisory committee meeting, with three women with prior GDM or HDP, to ensure adequate consumer feedback and the acceptability of the predictor variables for women. 

### 2.5. Data Analysis 

Data for online surveys were collected using Qualtrics (Qualtrics, Provo, UT, USA). The researchers developed and tested the survey questions for usability and face validity and piloted the questions with clinicians, researchers and women with prior GDM and HDP. The questions were updated and retested to ensure clarity. Participant demographics were collected to ensure adequate representation across disciplines. Results are reported using median, IQR and percentages. Statistical analyses were conducted using Microsoft^®^ Excel^®^ 2019 MSO (16.0.10402.20023) (Microsoft Corp., Redmond, WA, USA). 

### 2.6. Ethics Statement

Ethical approval was obtained from Monash Health Human Research Ethics Committee (Project Number 74671). Written informed consent was not collected due to the expert nature of the Delphi panel.

## 3. Results

### 3.1. E-Delphi Round One Online Survey 

The Delphi panel comprised 31 clinicians and academics (50% response rate) (Table 1). Workplace locations at the time of survey completion included Australia, Iran, Sweden, Norway and the United Kingdom. The majority of panelists were female (77%, n = 24). The participants were obstetricians (23%, n = 7), endocrinologists (19%, n = 6), midwives (16%, n = 5), dietitians (16%, n = 5), epidemiologists (10%, n = 3), biostatisticians (6%, n = 2), health economists (3%, n = 1) and general practitioners (3%, n = 1). One participant did not disclose their profession (n = 1).

After the first round, nine of the 15 predictor variables presented to the panelists (60%) were identified as important for inclusion in the risk prediction model, and six predictor variables (40%) were equivocal, with no clear consensus (Table 2). No predictor variables were of low importance. The predictor variables nominated for further consideration (n = 8) included history of diabetes (type one diabetes mellitus (T1DM), T2DM and GDM), history of polycystic ovarian syndrome, history of macrosomia, history of chronic kidney disease, waist circumference, postcode (or another measure of socioeconomic status), family history of diabetes (including T1DM, T2DM and GDM) and family history of hypertension (including chronic hypertension and HDP).

### 3.2. Consensus Meeting and Round Two Online Survey 

Thirty-nine percent (n = 12/31) of the panelists from *Round One* and two women with prior GDM participated in the consensus meeting (n = 14). The panelists included endocrinologists (26%, n = 4), epidemiologists (14%, n = 2), dietitians (14%, n = 2), women with prior GDM (14%, n = 2), biostatisticians (7%, n = 1), general practitioners (7%, n = 1), midwives (7%, n = 1) and obstetricians (7%, n = 1) (Appendix A). The panelists discussed the importance, feasibility and acceptability of the 23 nominated variables from *Round One*. The panelists suggested ways to optimize the definitions of predictor variables. Presenting ‘history of GDM’ as a separate predictor to ‘history of’ or ‘pre-existing diabetes (T1DM and T2DM)’ was recommended. The panelists could not agree on how to reflect blood glucose status within the risk prediction model, suggesting that HbA1c be accompanied by either a fasting or random blood glucose measure. The women with prior GDM highlighted that first-trimester blood tests are typically not performed under fasting conditions, and a random measure may be more acceptable. Providing clear definitions of what constitutes a history of chronic hypertension, chronic kidney disease and macrosomia was recommended to ensure consistency in interpretation by clinicians when using the tool. The panelists highlighted that the questions pertaining to and categorization of ethnicity/race data collection should be specific to the country/region, and the terminology should facilitate accurate interpretation by the patient. The women with prior GDM suggested that the predictor variables were not invasive, intensive or burdensome and that they were willing to undertake additional testing to mitigate the risk of pregnancy complications.

All fourteen panelists completed the *Round Two* survey. Fifteen of the 24 proposed predictor variables were selected for inclusion in the risk prediction tool (Figure 2), including demographic variables (age, ethnicity/race), pre-pregnancy history (body mass index, family history of diabetes, history of chronic kidney disease, history of polycystic ovarian syndrome, height, pre-existing diabetes, pre-existing hypertension), obstetric history (history of GDM, history of macrosomia, history of pre-eclampsia, parity), maternal biochemical measures (blood glucose levels) and maternal hemodynamic measures (systolic blood pressure). Nine predictor variables were excluded (Table 3).

## 4. Discussion

Fifteen predictor variables relating to maternal demographics, pre-pregnancy history, obstetric history, biochemical and hemodynamic risk factors were selected by experts and stakeholders to include in a novel, prognostic, composite risk prediction tool for GDM and HDP. The variables identified in an evidence synthesis were subjected to a two-round e-Delphi process with an integrated consensus-making workshop. Stakeholders were engaged to select clinically relevant predictor variables with predictive power for inclusion in the risk prediction tool. All selected predictor variables are commonly reported, well-established risk factors or routine measurements used to assess the risk of developing or diagnose HDP [2] and/or GDM [3].

The stakeholders were engaged to ensure that predictor variables were acceptable to collect in early pregnancy. Commonly reported barriers to the routine use of risk prediction tools in practice include lack of time during consultations, limited or an overwhelming number of tools or disdain towards prediction tools that are overly simple and seem to disregard clinical complexity [6,17]. Risk prediction tools should ideally use a small but meaningful number of predictor variables that are unambiguous, easy to measure, low cost, widely available and easy for healthcare providers and patients to understand [6]. A modified e-Delphi process enabled an in-depth discussion to confirm that the selected variables aligned with these considerations to improve the uptake of the final tool into routine antenatal care. 

Subsequent stages of tool development will include deriving a clinical prediction rule or model using the final fifteen variables. Using a retrospective cohort design, the prediction model will be internally validated using a large dataset for pregnancies resulting in birth from one of the largest maternity services in Australia. External validation, approaches to communicate the model results to women, implementation strategies and the implications of the risk prediction tool on treatment decision-making will need to be explored. 

Accurately predicting the risk of conditions that fall within the umbrella term of HDP may pose challenges. Pre-eclampsia is likely comprised of multiple subtypes, with varied phenotypic presentations [18]. Preterm pre-eclampsia is often predicted in the first trimester with abnormal uterine Doppler velocimetry, low pregnancy-associated PaPP-A or PlGF [19,20]. The panel excluded these predictor variables due to challenges with data collection in low-income, primary and/or community healthcare settings. The ability of the final risk prediction tool to accurately predict the varied phenotypic subtypes or conditions within HDP, such as preterm versus term pre-eclampsia, will need to be investigated.

The key strengths of this project were selecting variables from evidence synthesis, broad multidisciplinary engagement and using a modified e-Delphi process to meaningfully involve stakeholders in the initial tool development. The modified e-Delphi process was efficient, with only two rounds of surveys needed to reach a consensus, consistent with many other Delphi studies [21]. Researchers are recommended to undertake a predictor-finding study to aid the identification of relevant variables when developing a risk prediction tool [7]. We conducted a systematic search and narrative review [9]. However, evaluating model characteristics was outside the review’s scope and a limitation of the review.

A modified e-Delphi process combines multiple anonymous surveys with an online workshop. Although anonymous online surveys encourage honesty and independent contribution from panelists [11,22], it is challenging to explore differing perspectives and the breadth of ideas within surveys alone [23]. The Delphi process was modified to include a workshop to facilitate in-depth, rich discussions between experts. The panelists were asked to anonymously vote on predictor variables for inclusion in the model to prevent groupthink or prevent experts from conforming to a dominant view within a group context [22]. 

The Delphi panel sample size was moderate, with 31 and 14 panelists in *Rounds One and Two*, respectively. While the optimal size of a Delphi panel is not known, a representative panel was prioritized. A panel of 10–20 participants for *Round Two* was prioritized to support meaningful, in-depth and timely workshop discussions. The initial response rate of only 50% for *Round One* and the high attrition rate between *Rounds One and Two*, likely due to the practical challenges of attending a live online consensus meeting, was a limitation of this study. Australian and international subject experts discussed the relevance of the predictor variables across multiple countries. The participants were primarily from higher-income countries. Further investigation into the feasibility of collecting predictor variables in low-income healthcare settings is warranted. The ranking of variables based on their predictive power in predicting GDM and/or HDP was not completed as part of this study and will be investigated in the development of the risk prediction model. Finally, a sentiment and dissent analysis, commonly used to explore experts’ confidence in their expertise on the topic and the level of agreement between experts, was not completed and is a limitation of this study [24].

## 5. Conclusions

This study describes the successful use of a modified e-Delphi process to engage stakeholders to collaboratively identify and select clinically important predictor variables in predicting the risk of GDM and HDP. The Delphi process efficiently consolidated expert opinion. The final fifteen predictor variables selected by the expert panel will inform the next phase of the risk prediction tool development. Engaging stakeholders in the initial phase of the development of a composite, prognostic, cardiometabolic risk prediction tool will support the development of a relevant and robust tool for implementation and translation into routine antenatal care.

## Figures and Tables

**Figure 1 healthcare-12-01361-f001:**
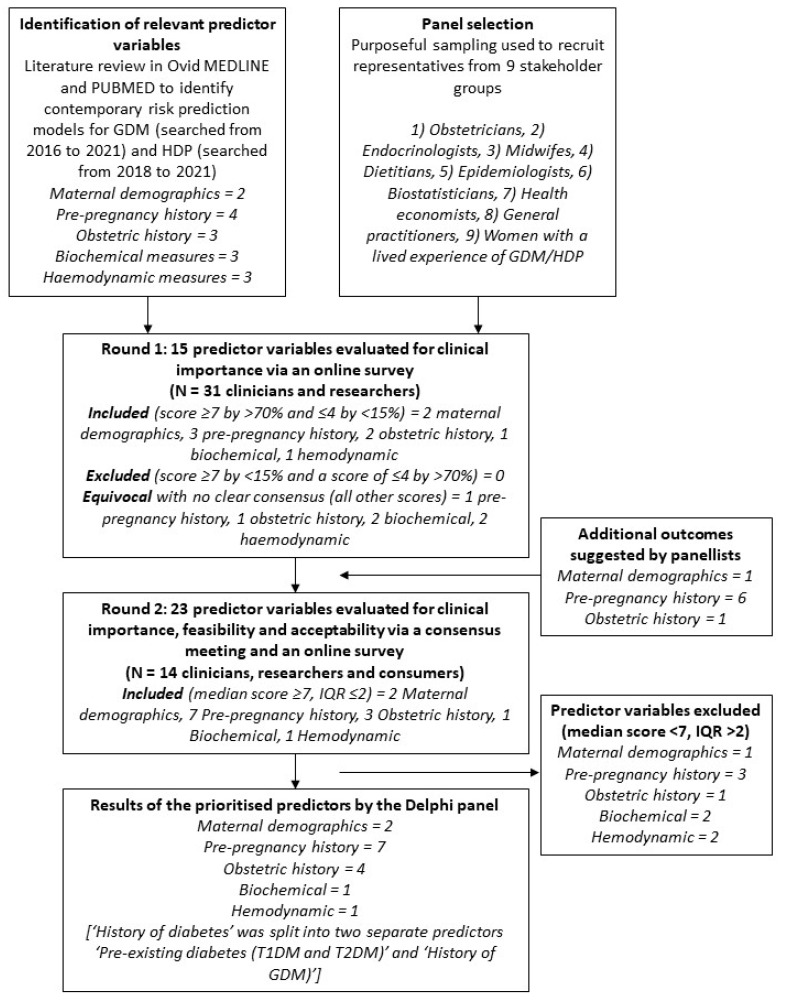
Flow chart of study design and outcome selection. Abbreviations: GDM, gestational diabetes mellitus; HDP, hypertensive disorders of pregnancy; T1DM, type one diabetes mellitus; T2DM, type two diabetes mellitus.

**Figure 2 healthcare-12-01361-f002:**
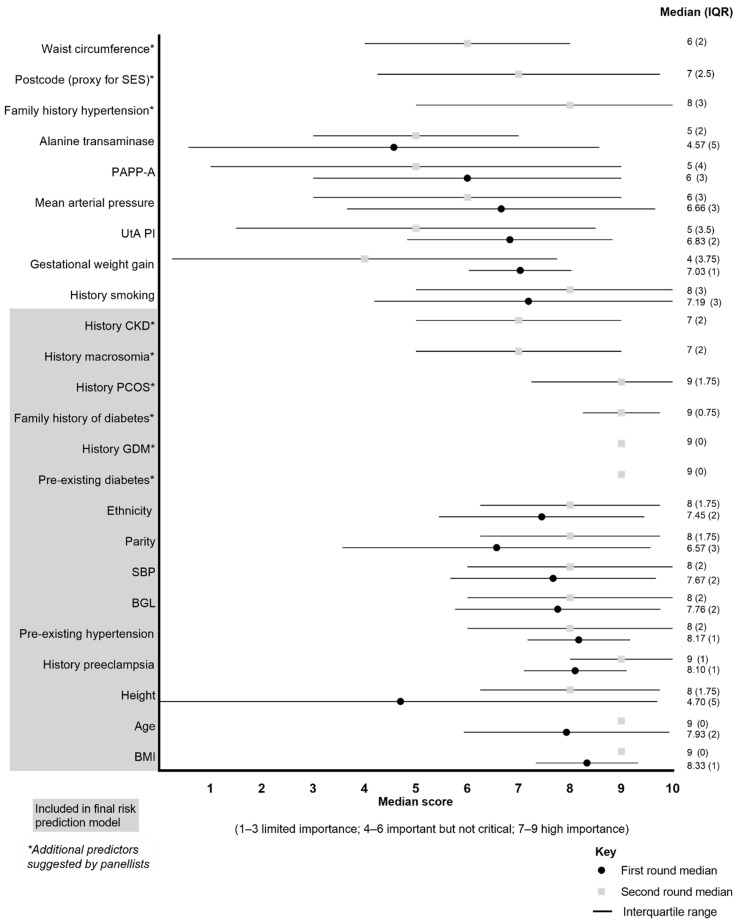
Prioritization of predictor variables during the first and second rounds. Abbreviations: BGL, blood glucose level; BMI, body mass index; CKD, chronic kidney disease; GDM, gestational diabetes mellitus; IQR, interquartile range. PAPP-A, pregnancy-associated plasma protein A; PCOS, polycystic ovary syndrome; SBP, systolic blood pressure; SES, socioeconomic status; UtA PI, uterine artery pulsatility index. For the second-round rating of predictors, acceptability and feasibility for collecting measures in early pregnancy were also considered.

**Table 1 healthcare-12-01361-t001:** Demographic characteristics of Delphi participants.

Participant Characteristic ^a^	*Round One* (n = 31)	*Round Two* (n = 14)
Age, mean ± SD	42 ± 10	43 ± 10
Gender (Female)	77 (24)	86 (12)
Profession		
Obstetrics and gynecology	23 (7)	7 (1)
Endocrinologist	19 (6)	29 (4)
Midwife	16 (5)	7 (1)
Dietitian	16 (5)	14 (2)
Epidemiologist	10 (3)	14 (2)
Biostatistician	6 (2)	7 (1)
Health economist	3 (1)	0
General practitioner	3 (1)	7 (1)
Consumer with a lived experience of gestational diabetes mellitus	0	14 (2)
Did not disclose profession	3 (1)	0

^a^ Data are presented as % (n) unless otherwise specified.

**Table 2 healthcare-12-01361-t002:** Results of the *Round One* Delphi survey.

Outcome	% of Panelists Ranking Each Outcome as Important (Score > 7/10)
Outcomes with consensus in ^a^	
Body mass index	90
Age	84
Pre-existing (chronic) hypertension	84
Ethnicity	81
History pre-eclampsia	81
Blood glucose level	81
Systolic blood pressure	81
History of smoking	74
Gestational weight gain	74
Outcomes with consensus out ^b^	
Nil	-
Outcomes with no consensus ^c^	
Mean arterial pressure	55
Parity	45
Uterine artery pulsatility index	39
Pregnancy-associated plasma protein A	32
Height	26
Alanine transaminase	19
Additional variables nominated for consideration	
Family history of diabetes (including T1DM, T2DM and GDM)	
Family history of hypertension (including chronic hypertension and HDP)	
History of chronic kidney disease	
History of diabetes (including T1DM, T2DM and GDM) ^d^	
History of macrosomia	
History of PCOS	
Postcode (or another measure of socioeconomic status)	
Waist circumference	

^a^ Outcomes with consensus In: score of ≥7 by more than 70% of participants and a score of ≤4 by <15% of participants. ^b^ Outcomes with consensus Out: score of ≥7 by <15% of participants and a score of ≤4 by more than 70% of participants. ^c^ Outcomes with no consensus: any other score combinations. ^d^ This variable was split into two unique variables during the online consensus meeting: pre-existing diabetes (including T1DM, T2DM) and history of gestational diabetes. Abbreviations: GDM, gestational diabetes mellitus; HDP, hypertensive disorders of pregnancy; PCOS, polycystic ovarian syndrome; T1DM, type one diabetes mellitus; T2DM, type two diabetes mellitus.

**Table 3 healthcare-12-01361-t003:** Reasons for predictor variable exclusion.

Variable	Reason for Exclusion
Gestational weight gain	A poor measure of risk in early pregnancy (most women do not gain excessive weight in the first 12 weeks). Difficult to accurately measure (pre-pregnancy weight at the first antenatal appointment is often self-reported).
Uterine artery pulsatility index	Access to appropriate equipment and training limits universal uptake in resource-poor healthcare settings.
Alanine transaminase and pregnancy-associated plasma protein A	Viewed as disease markers rather than risk factors. Not always routinely collected in early pregnancy. There are potential issues surrounding cost, either to the individual and/or healthcare system.
Postcode	Postcode was proposed as a proxy measure for socioeconomic status. While capturing socioeconomic status was viewed as important, this variable limited the transferability of the tool to all countries/regions.
Waist circumference and mean arterial pressure	BMI and systolic blood pressure are substitute measures that can be easily collected with little training required.
History of smoking	Viewed as clinically relevant to HDP though not GDM (when considering pathophysiological pathways).
Family history of hypertension (including chronic hypertension and HDP)	Relies on self-reported data that is difficult to accurately collect.

Abbreviations: BMI, body mass index; GDM, gestational diabetes mellitus; HDP, hypertensive disorders of pregnancy.

## Data Availability

The participants of this study did not give written consent for their data to be shared publicly. Supporting data are not available due to the nature of this research and potential for re-identification.

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
