# Peer review of "Identifying Predictor Variables for a Composite Risk Prediction Tool for Gestational Diabetes and Hypertensive Disorders of Pregnancy: A Modified Delphi Study"

_healthcare, 2024, doi:10.3390/healthcare12131361_

Round 1

Reviewer 1 Report

Comments and Suggestions for Authors

Interesting 'global' exercise, following a well-established procedural model.

Comments/remarks:

- As the world 'atomizes', efforts to construct global prediction tools will yield diminishing returns on investment.  At the institution of this reviewer, screening policies for pre-eclampsia and GDM are well in place.  The authors may have 'ditched' the mean arterial pressure and uterine artery pulsatility index as relevant outcome variables, but the current "no consensus" is unlikely to change the minds of many prudent clinicians.

- In the Discussion, the authors admit that hypertensive disease in pregnancy is an umbrella term.  All students are taught the primordial difference between early and late pre-eclampsia (and the role of spiral artery remodeling gone wrong vs. villous crowding) and the diverse pathophysiology underlying a clinical diagnosis of "gestational hypertension" (either non-proteinuric PE or latent (essental) hypertension).  The higher the obesity prevalence in a certain area, the higher the prevalence of the second pathophysiology.  The danger exists that we are making tools based on (and only relevant for) a world with a high obesity prevalence.

- The same is true for GDM.  In most cases, as we all know, the risk factors are identical to type 2 diabetes and essential hypertension later in life.  But again, the 'losers' are non-obese gravidas with GDM owing to a restrained beta-cell capacity.  These women may (quite unexpectedly according to the model) develop a fulminant type 1 diabetes in the first postpartum year (and are at risk for keto-acidosis throughout).  Auto-antibodies and T1DM screening apparently have no place in a global obesity world.  Obesity is the ever-present signal, the rest is noise.  Again, many prudent clinicians in Northern Europe (which sees a higher prevalence of T1DM) will hesitate to find patient-centered guidance in the proposed "composite risk prediction tool". 

Lines 241-2: "ethnicity/race data collection should be specific to the country/region".  Fair enough, this is a sensitive topic for a global tool.  Nevertheless, there is a large body of scientific literature showing that people originating from the Middle East and South Asia, for example, have a higher level of insulin resistance controlled for BMI.  It is in the best interest of all people involved that the science can prevail.    

Author Response

Dear Reviewer,

Re: Identifying predictor variables for a composite risk prediction tool for Gestational Diabetes and Hypertensive Disorders of Pregnancy: a modified Delphi study (Manuscript ID: healthcare-3058597)

Thank you for offering to publish our manuscript with minor revisions. We thank the reviewers for their time and feedback on our manuscript.

We have addressed all reviewer comments. The tables below summarise how we updated the manuscript in response to each reviewer's comment.

We have also added additional information to the funding details in track changes.

Please let us know if you would require any further revisions or information.

Thank you.

Kind regards,

Dr. Sarah Lang and Dr. Stephanie Cowan

Monash Centre for Health Research and Implementation (MCHRI)

Reviewer 1 – Comments

Reviewer Comment

Author Response

Interesting 'global' exercise, following a well-established procedural model.

Comments/remarks:

As the world 'atomizes', efforts to construct global prediction tools will yield diminishing returns on investment. At the institution of this reviewer, screening policies for pre-eclampsia and GDM are well in place. The authors may have 'ditched' the mean arterial pressure and uterine artery pulsatility index as relevant outcome variables, but the current "no consensus" is unlikely to change the minds of many prudent clinicians.

Thank you for your comments. We agree that arterial pressure and uterine artery pulsatility index are relevant and important screening measures. However, due to the burden and challenges associated with collection of these measures, experts excluded these measures to improve the acceptability and feasibility of implementing the final tool in varied healthcare contexts.

In the Discussion, the authors admit that hypertensive disease in pregnancy is an umbrella term. All students are taught the primordial difference between early and late pre-eclampsia (and the role of spiral artery remodeling gone wrong vs. villous crowding) and the diverse pathophysiology underlying a clinical diagnosis of "gestational hypertension" (either non-proteinuric PE or latent (essental) hypertension). The higher the obesity prevalence in a certain area, the higher the prevalence of the second pathophysiology. The danger exists that we are making tools based on (and only relevant for) a world with a high obesity prevalence.

The same is true for GDM. In most cases, as we all know, the risk factors are identical to type 2 diabetes and essential hypertension later in life. But again, the 'losers' are non-obese gravidas with GDM owing to a restrained beta-cell capacity. These women may (quite unexpectedly according to the model) develop a fulminant type 1 diabetes in the first postpartum year (and are at risk for keto-acidosis throughout). Auto-antibodies and T1DM screening apparently have no place in a global obesity world. Obesity is the ever-present signal, the rest is noise. Again, many prudent clinicians in Northern Europe (which sees a higher prevalence of T1DM) will hesitate to find patient-centered guidance in the proposed "composite risk prediction tool".

Thank you for your comments and reflection on the challenges with risk prediction for gestational diabetes and hypertensive disorders of pregnancy.

Lines 241-2: "ethnicity/race data collection should be specific to the country/region". Fair enough, this is a sensitive topic for a global tool. Nevertheless, there is a large body of scientific literature showing that people originating from the Middle East and South Asia, for example, have a higher level of insulin resistance controlled for BMI. It is in the best interest of all people involved that the science can prevail.

Thank you. We agree that ethnicity/ race is an important consideration when predicting the risk of Gestational Diabetes and Hypertensive Disorders of Pregnancy. We have updated the sentence to specify, “Panelists highlighted that the questions pertaining to and the categorization of ethnicity/race data collection should be specific to the country/region, and the terminology should facilitate accurate interpretation by the patient.”

Reviewer 2 Report

Comments and Suggestions for Authors

The authors took an intermediate step in the development of a combined predictive model for gestational diabetes and hypertensive disorders of pregnancy.  They took prior work that developed a list of risk factors for the combined diseases and by way of a modified two-step Delphi study reached a consensus on a list of variables that were predictive, acceptable in first trimester exams, and feasible in practice.

The report is well developed and well written.  It remains to be seen if this more protracted effort yields a model that is significantly superior to the approximately 34 previous models for either gestational diabetes or hypertensive disorders of pregnancy.

Abstract

Line 18: “Formative research with a systematic review” suggests to the casual reader that the authors undertook a systematic review in this paper.  Possibly try, “A recently performed systematic review” or simply, “A prior systematic review”.

Lines 22-23: The sentence starting, “Overall..” is confusing since the beginning and ending number of variables is the same.  This suggests that the entire Delphi process was unnecessary.  It would be helpful to indicate that 9 additional variables were submitted by the Round One online meeting.

Line 28: The authors are clearly preparing to create a combined prediction model for GDM and HDP.  It is important to make clear to the casual reader that the model was not created in this paper.  To that end, I found the word “prioritized” a bit troubling here.  The variables were only really collected or listed.  They were not prioritized from most important to least important.

Discussion

Limitations: This lack of ranking or prioritizing appears to me to be a limitation of this study.  This can be made clear here or the authors may suggest that the work of ranking the variable would be part of the next step, the actual development of the prediction model.

Author Response

Dear Reviewer,

Re: Identifying predictor variables for a composite risk prediction tool for Gestational Diabetes and Hypertensive Disorders of Pregnancy: a modified Delphi study (Manuscript ID: healthcare-3058597)

Thank you for offering to publish our manuscript with minor revisions. We thank you for your time and feedback on our manuscript.

We have addressed all comments. The tables below summarise how we updated the manuscript in response to each comment.

We have also added additional information to the funding details in track changes.

Please let us know if you would require any further revisions or information.

Thank you.

Kind regards,

Dr. Sarah Lang and Dr. Stephanie Cowan

Monash Centre for Health Research and Implementation (MCHRI)

Reviewer 2 – Comments

Reviewer Comment

Author Response

The authors took an intermediate step in the development of a combined predictive model for gestational diabetes and hypertensive disorders of pregnancy. They took prior work that developed a list of risk factors for the combined diseases and by way of a modified two-step Delphi study reached a consensus on a list of variables that were predictive, acceptable in first trimester exams, and feasible in practice.

The report is well developed and well written. It remains to be seen if this more protracted effort yields a model that is significantly superior to the approximately 34 previous models for either gestational diabetes or hypertensive disorders of pregnancy.

Thank you for your comments and feedback.

Abstract

Line 18: “Formative research with a systematic review” suggests to the casual reader that the authors undertook a systematic review in this paper. Possibly try, “A recently performed systematic review” or simply, “A prior systematic review”.

Thank you. This line and the methods section has been updated to state, “A prior systematic review” for clarity.

Lines 22-23: The sentence starting, “Overall..” is confusing since the beginning and ending number of variables is the same. This suggests that the entire Delphi process was unnecessary. It would be helpful to indicate that 9 additional variables were submitted by the Round One online meeting.

Thank you. This line has been updated to state that multidisciplinary experts “nominated eight additional variables for consideration” for clarity.

Line 28: The authors are clearly preparing to create a combined prediction model for GDM and HDP. It is important to make clear to the casual reader that the model was not created in this paper. 

To that end, I found the word “prioritized” a bit troubling here. The variables were only really collected or listed. They were not prioritized from most important to least important.

Thank you. A sentence has been added to the abstract highlighting that the development of the model will be explored in a separate publication, “Variables will inform the development of a cardiometabolic risk prediction tool in subsequent research.”

The word prioritize has been changed to select.

Discussion

Limitations: This lack of ranking or prioritizing appears to me to be a limitation of this study. This can be made clear here or the authors may suggest that the work of ranking the variable would be part of the next step, the actual development of the prediction model.

Thank you. We have added a sentence to the study limitations, “Ranking of variables based on their predictive power in predicting GDM and/or HDP was not completed as part of this study and will be investigated in the development of the risk prediction model.”
